# Dopant-Free Hole Transport Materials with a Long Alkyl Chain for Stable Perovskite Solar Cells

**DOI:** 10.3390/nano9070935

**Published:** 2019-06-28

**Authors:** Kai Wang, Haoran Chen, Tingting Niu, Shan Wang, Xiao Guo, Hong Wang

**Affiliations:** 1Key Laboratory of Flexible Electronics (KLOFE) & Institute of Advanced Materials (IAM), Jiangsu National Synergetic Innovation Center for Advanced Materials (SICAM), Nanjing Tech University (Nanjing Tech), Nanjing 211816, China; 2Zhongshan Institute of Modern industrial Technology, South China University of Technology, Zhongshan 528437, China

**Keywords:** perovskite solar cell, alkyl chain, hole transporting materials, stable

## Abstract

Hole transport materials are indispensable to high efficiency perovskite solar cells. Two new hole transporting materials (HTMs), named 4,4′-(9-nonyl-9H-carbazole-3,6-diyl)bis (*N*,*N*-bis(4-methoxyphenyl)aniline) (CZTPA-1) and 4,4′-(9-methyl-9H-carbazole-3,6-diyl)bis (*N*,*N*-bis(4-methoxyphenyl)aniline)(CZTPA-2), were developed by different alkyl substitution methods. The two compounds, containing a carbazole core and triphenylamine (TPA) groups with different lengths of the alkyl chain, were designed and synthesized through a two-step synthesis approach. The power conversion efficiency (PCE) was found to be affected by the length of the alkyl chain, reaching 7% for CZTPA-1 and 11% for CZTPA-2. Furthermore, the CZTPA-2 still maintained 89.7% of its original performance after 400 h. The proposed results demonstrate the effect of carbon chain substituents on the efficiency of perovskite solar cells (PSCs).

## 1. Introduction

Perovskite solar cells (PSCs) have attracted much attention in recent years owing to their excellent optoelectronic properties, high absorption coefficients, easy solution processability, high carrier mobility, and so on [1,2,3,4,5]. Many efforts have been made to increase the efficiency of PSCs. Surprisingly, it has increased dramatically from 3.8% [1] to 24.2% [6] in a short period of time.

As is well-known, 2,2′,7,7′-Tetrakis[*N*,*N*-di(4-methoxyphenyl)amino]-9,9′-spirobifluorene (Spiro-OMeTAD) [7,8] is a common hole transporting material (HTM) with good performance. However, it is still a formidable challenge to develop this material due to various drawbacks, such as a complicated synthesis process and high cost. Therefore, new HTMs with a simple synthesis process and low cost need urgently to be developed.

HTMs construct the hole transport layer (HTL) that is needed to block electron transport, enhance hole transport, and prevent direct contact between the perovskite layer and the electrode, which causes annihilation. An ideal HTM should have such characteristics as good hole mobility, good hydrophobicity, suitable energy levels, and the ability to be prepared in solution [9,10,11]. Currently, common HTMs can be classified as inorganic substances, organic polymers, or organic small molecules depending on the type of material. Low cost, high stability, and hole mobility are among the many advantages that HTMs based on inorganic materials have. Kamat and co-workers first introduced copper iodide as an HTM and achieved an average PCE of 6% [12]. Then, inorganic semiconductors (CuSCN, NiO, CuI) as HTMs were developed [13,14]. In addition to inorganic HTMs, polymer-based HTMs have been explored in PSCs. Poly-[bis(4-phenyl)(2,4,6-trimethylphenyl)amine] (PTAA) was the first conjugated polymer to be used as an HTM in PSCs [15] and maintained the highest PCE of any reported polymeric HTM [16]. Accompanied by the development of conjugated polymer HTMs, Poly(3-hexylthiophene-2,5-diyl (P3HT) [17] was originally used in organic solar cells (OSCs) as the active layer was introduced into the hole transport layer. Compared to polymeric HTMs, small molecule HTMs have the advantages of a determined molecular weight and simple purification for PSCs. As a group with good stability and solubility, triphenylamine (TPA) is widely used in organic small molecule HTMs [18,19,20]. The TPA group in particular can influence the optoelectrical properties of HTMs due to its non-planar geometry. Furthermore, the carbazole group has a high carrier mobility in OSCs and as good a performance as HTMs; thus, it is currently regarded as a core in PSCs [21,22,23,24,25,26,27].

In 2014, Sun and co-workers [27] designed a series of HTMs based on carbazole, named X19 and X51. The X51 realized a PCE of 9.8% due to a higher charge-carrier mobility and conductivity than X19. Subsequently, Nazeeruddin and co-workers [28] studied bridged carbazole with biphenyl, by using silolothiophene as the bridge, which obtained a PCE of 13.1%. They found that compared to the spirofluorene-linked triphenylamine HTMs, novel silolothiophene-linked methoxy triphenylamines (Si-OMeTPAs) enable more stable PSCs. In the same year, Tang and co-workers [29] used carbazole as a core with four TPAs as the side groups to achieve a PCE of up to 18.32%.

Recently, Khaja Nazeeruddin and co-workers [30] used anthra[1,2-b:4,3-b′:5,6-b′:8,7-b‴]tetrathiophene as the core, by changing the alkyl chain length of the methoxy groups on the triarylamine sites, to develop a series of materials. The device based on a methyl substitute named ATT-OMe was found to have the best PCE of 18.13%, which was better than that of the device based on other longer alkyl chain HTMs. They believed that the presence of alkyl chains decreased the hole-transport properties. However, the performance of PSCs based on carbazole is far from that of the classical Spiro-OMeTAD. Therefore, more efforts are needed to develop new small molecule HTMs that match with perovskite instead of the Spiro-OMeTAD.

In this work, we developed two HTMs—4,4′-(9-nonyl-9H-carbazole-3,6-diyl)bis (*N*,*N*-bis(4-methoxyphenyl)aniline) (CZTPA-1) and 4,4′-(9-methyl-9H-carbazole-3,6-diyl)bis (*N*,*N*-bis(4-methoxyphenyl)aniline) (CZTPA-2)—based on TPA as the end group. They both have the advantage of simple synthesis steps and low cost. Both HTMs are synthesized by one-step Suzuki coupling. The cost of the raw material 3,6-Dibromocarbazole ($0.3/g) plus 4-methoxy-*N*-(4-methoxyphenyl)-*N*-(4-(4,4,5,5-tetramethyl-1,3,2-dioxaborolan-2-yl)phenyl)aniline ($15/g) is clearly lower than that of Spiro-OMeTAD ($220/g). Notably, the CZTPA-2 with the longer alkyl chain achieved a better PCE of 11.79% with a short current density (*J*_sc_) of 21.80 mA/cm^2^, an open circuit voltage (*V*_oc_) of 0.99 V, and a fill factor (FF) of 54.59%. This is attributed to the significantly improved hole mobility of CZTPA-2, resulting in a significant increase in device efficiency.

## 2. Materials and Methods

### 2.1. Materials

Unless otherwise noted, all reagents used in the experiments were purchased from commercial sources and used without further purification. 3,6-Dibromo-9H-carbazole, *N*,*N*-bis(4-Methoxyphenyl)-4-(4,4,5,5-tetraMethyl-1,3,2-dioxaborolan-2-yl)-BenzenaMine, lead iodide (PbI_2_), methylammonium iodide (MAI), acetonitrile (99.8%), chlorobenzene (99.9%), and dimethylformamide (DMF) (99%) were purchased from Sigma-Aldrich. 4-tert-butylpyridine (TBP) and Li-bis-(trifluoromethanesulfonyl) imide (Li-TFSI) were purchased from TCI. 2,2′,7,7′-tetrakis-(*N*,*N*-di-p-methoxyphenylamine)-9,9’-spirobifluorene (Spiro-OMeTAD) (99.0%) was purchased from Xi’an Polymer Light Technology Co., Ltd.

Perovskite precursor: The perovskite precursor was obtained by mixing PbI_2_ and MACl (in a molar ratio of 1:1) in DMF with a concentration of 350 mg/mL, and was then stirred at 60 °C overnight in a glovebox.

Spiro-OMeTAD: The 2,2′,7,7′-Tetrakis(*N*,*N*’-di-p-methoxyphenylamine)-9,9’-spirobifluorene (Spiro-OMeTAD) was doped with TBP and Li-TFSI. A total of 73.2 mg of Spiro-OMeTAD (Xi’an Polymer Light Technology Co., Ltd., Xi’an, China) was dissolved in 1 mL of chlorobenzene (CB) with 28.8 μL of 4-tert-butylpyridine (TBP) and 17.6 μL of Li-bis-(trifluoromethanesulfonyl) imide (Li-TFSI).

### 2.2. Device Fabrication

The SnO_2_ layer was spin-coated on an ITO substrate at 3000 rpm for 30 s, which was cleaned in UV–ozone and then annealed at 150 °C for 30 min. The perovskite layer was spin-coated at 4000 rpm for 30 s by an anti-solvent method. The details of the operation are as follows: 100 μL CB was rapidly added after 5 s of spin-coating with perovskite solution; and the perovskite films were annealed at 100 °C for 5 min. All of the above processes were performed in the nitrogen glovebox. Then, Spiro-OMeTAD and two HTMs were dissolved in the CB (10 mg/mL), and then spin-coated upon the perovskite layer at 3000 rpm for 30 s. Then, molybdenum trioxide (MoO_3_) and gold (Au) were thermally evaporated on the hole transporting layer. The effective area of the cell is 0.05 cm^2^.

### 2.3. Device Characterization

The cross-sectional images of PSC were taken by scanning electron microscopy (SEM) (ZEISS Merlin, Carl Zeiss Microscopy, Jena, Germany). The UV–visible absorption spectra were measured using a UV Spectrophotometer (SHIMADZU UV-1750, East Test Technology Co., Ltd, Shenzhen, China). Photoluminescence (PL) spectra were obtained using a spectrofluorometer (HitachiF-7000, Hitachi High-Technologies Corporation, Shenzhen, China). Thermogravimetric (TGA) analysis was performed on a Mettler Toledo TGA2.

The device was measured under AM 1.5 G solar irradiation with an intensity of 100 mW/cm^2^ through an Enlitech SS-F5-3A solar simulator. The instrument was calibrated on standard solar cells. *J–V* properties were measured by the method of Enlitech Ltd. (Kaohsiung, Taiwan) and a Keithley (Cleveland, OH, USA) 2400 source meter under dark conditions. The external quantum efficiency (EQE) spectra were measured using a solar cell IPCE test system (CROWNTECH Inc., model QTEST HIFINITY (Macungie, PA, USA)).

The synthesis method and details of the experimental procedure are shown in the supporting information (SI).

## 3. Results

The details of the experimental procedure are shown in the Appendix A. The design principle was to improve planarity as well as increase solubility. The carbazole group with simple structures has good hole transporting ability. By inserting N atoms with different alkyl chain lengths, the optoelectronic properties and solubility can be adjusted. Herein, we chose 4-methoxy-*N*-(4-methoxyphenyl)-*N*-(4-(4,4,5,5-tetramethyl-1,3,2-dioxaborolan-2-yl)phenyl)aniline units as the raw material, used Suzuki coupling, and substituted the carbazole core with different lengths of alkyl chains. The aimed-for materials were obtained by simple column chromatography separation and recrystallization.

Figure 1a shows the absorption spectra of CZTPA-1 and CZTPA-2 in chloroform and as a coating on quartz substrates. Absorption peaks at 335 nm for CZTPA-1 and 327 nm for CZTPA-2 in the solution were observed. Relative to the solution, the CZTPA-1 film exhibits a redshift of 2 nm with an onset of 409 nm, corresponding to an optical bandgap of 3.03 eV, whereas CZTPA-2 aligns to a narrower bandgap of 2.98 eV (onset of 416 nm). Both HTMs have a slight redshift in the film compared with the solution, suggesting that aggregation exists in the films.

From Figure 1b, the highest occupied molecular orbital (HOMO) level of CZTPA-1 is −4.89 eV, while in the CZTPA-2 there is a higher HOMO of −4.83 eV (the energy level of Ferroceneis is −0.54 eV by the cyclic voltammetry method). Compared to the two HTMs, the Spiro-OMeTAD has the higher energy level as the HOMO is −4.64 eV. Based on the relationship of *E*_LUMO_ = *E*_HOMO_ + *E*_g_, the lowest unoccupied molecular orbital (LUMO) level value was calculated to be −1.86 eV for CZTPA-1, −1.85 eV for CZTPA-2, and −1.74 eV for Spiro-OMeTAD (Table 1). CZTPA-2 has similar electrochemical properties compared with CZTPA-1, which indicates that the electrochemical properties of the molecule remain stable with a change in the alkyl chain length.

We then found that CZTPA-1 and CZTPA-2 match well with the energy level of the perovskite from Figure 2a. Furthermore, the thermal stability of the two HTMs was measured by thermogravimetric analysis (TGA) (Figure 2b). Both CZTPA-1 and CZTPA-2 exhibit good thermal stability with decomposition temperatures (T_d_, 5% weight loss) at 391.8 °C and 384.6 °C, respectively. As we expected, increasing the length of the alkyl chains reduces the thermal stability.

The photoluminescence (PL) spectra in Figure 2c show the maximum emission peak at 431 nm for CZTPA-1 and at 425 nm for CZTPA-2 in film. CZTPA-2 was slightly more blue-shifted than CZTPA-1, caused by the increase in the length of the alkyl chains.

Figure 2d shows the PL spectra of the perovskite, perovskite with Spiro-OMeTAD, perovskite with CZTPA-1, and perovskite with CZTPA-2. Strong PL quenching was observed after the HTMs were coated on perovskite films. Respectively, compared with the original perovskite film, the PL intensity was reduced to 7%, 22%, and 18% after coating with Spiro-OMeTAD, CZTPA-1, and CZTPA-2. Thus, we think that CZTPA-2 has better charge separation than CZTPA-1 and a smaller *J_sc_* and FF in the PSCs compared to Spiro-OMeTAD devices. In short, this means that the hole transfer capabilities of CZTPA-2 are superior because of their better charge transfer capability.

Figure 3 shows two cross-sectional scanning electron microscopy (SEM) images of the PSC with the structure of ITO/SnO_2_/perovskite/HTMs/Au. The PSC includes an ≈460 nm perovskite capping layer and an ≈35 nm HTM layer (CZTPA-1 or CZTPA-2). From the SEM image, we found that the HTM layer deposited on the perovskite layer and the boundaries of each layer are clear. Atomic force microscopy (AFM) images of the two materials spin-coated onto perovskite films are shown in Appendix A, which represent the perovskite/CZTPA-2 (a) and the perovskite/CZTPA-1 (b) films, respectively. The roughness of both samples is slightly high, which may be due to the lower thickness of the HTMs. After a comparison, it was found that the roughness of the film in which the HTM is CZTPA-2 (RMS = 20.968 nm) is significantly lower than that of CZTPA-1 (RMS = 28.662 nm); this is attributed to the solubility of CZTPA-2 being higher such that it could better cover the film and further improve the carrier transport of the device.

In order to compare the properties of the two HTMs, we used them as the hole transport layer of perovskite solar cells to compare device performance. Between them, the effective area of the cells is 0.05 cm^2^, and the scanning rate is 0.02 V/s. As shown in Figure 4a,b, the PSCs with CZTPA-2 achieve a PCE of 11.79% with an open-circuit voltage (*V*_oc_) of 0.99 V, a short-circuit current density (*J*_sc_) of 21.8 mA cm^−2^, and a fill factor (FF) of 54.59%, while the PSCs with CZTPA-1 achieve a lower PCE of 6.05% under the condition of no doping. In contrast, the best cell is based on Spiro-OMeTAD, as the HTM achieves the PCE of 16.77%. We also compared the Spiro-OMeTAD PSC without any doping, which achieved a PCE of 11.74%. The photovoltaic performance of the PSCs based on the two HTMs (dopant-free) and the Spiro-OMeTAD were investigated under AM 1.5 illumination (100 mW cm^−2^). The champion device performance plots of CZTPA-1 and CZTPA-2 are shown in Table 2. The device fabricated with CZTPA-1 as the HTM yielded a promising PCE of 6.05% with a *J*_sc_ of 20.58 mA cm^−2^, a *V*_oc_ of 0.77 V, and an FF of 38.01%. We calculated the average PCE of the CZTPA-2 devices to be 10.15% ± 0.90. The CZTPA-1 devices own an average PCE of 5.27% ± 0.57. The average PCE of the Spiro-OMeTAD (dopant-free) is 10.02% ± 0.98, and the corresponding PCE of Spiro-OMeTAD is 15.65% ± 0.71. All data are based on the values obtained from 20 devices. From these, we can speculate that CZTPA-2 exhibits better performance compared with CZTPA-1. The FF and *V*_oc_ of CZTPA-1 are obviously lower than those of CZTPA-2 in the PSC devices. Furthermore, CZTPA-2 obtains a slightly higher PCE compared with the dopant-free Spiro-OMeTAD, which is attributed to the higher *J*_sc_.

When preparing the solution, we found the CZTPA-1 dissolution rate to be slower than the CZTPA-2 dissolution rate. What is more, the solubility of CZTPA-2 was found to be better than that of CZTPA-1 with an increasing alkyl chain length and CZTPA-2 deposits in perovskite with better crystallization. These results also demonstrate that CZTPA-2 achieves better performance.

To test the reproducibility of the devices, we fabricated 20 devices in several different batches. The devices are shown in Figure 4c,d. As shown in the PCE histogram of the corresponding device data, the average PCE of CZTPA-2 and CZTPA-1 is 10.15% and 5.27%, respectively.

Moreover, the EQE spectrum of PSCs with CZTPA-2 is also shown in Figure 4b. The integral of the current densities calculated from the EQE spectra is 21.32 mA cm^−2^ for CZTPA-2 according predominantly to the experimental data.

To calculate the hole mobility of the two HTMs, we constructed a device with a configuration of ITO/PEDOT:PSS/HTM/Au using the space-charge-limited current (SCLC) method, and the *J–V* characteristics of this device were studied in the dark. Hole mobility is calculated by the Mott–Gurney equation of *J* = 9ε_r_ε_0_μVa^2^/8L^3^, so we change the form of the formula to obtain μ = 8d^3^/9ε_r_ε_0_(*J*^1/2^/Va)^2^, where ε_r_ is the relative dielectric constant of the transport medium (ε_r_ = 3 for organic materials), ε_0_ is the permittivity of free space (8.85 × 10^−12^ C V^−1^ m^−1^), *J* is the dark current density (mA cm^−2^), and d is the thickness of the active layer [31]. d is 48 nm for CZTPA-1 and 56 nm for CZTPA-2. Figure 5 shows that the hole mobility of Spiro-OMeTAD (doped) is 1.01 × 10^−3^ cm^2^ V^−1^ s^−1^. The hole mobility of CZTPA-1 is 4.68 × 10^−5^ cm^2^ V^−1^ s^−1^, and CZTPA-2 has the higher hole mobility of 8.06 × 10^−5^ cm^2^ V^−1^ s^−1^. However, they are all lower than the hole mobility of Spiro-OMeTAD (doped). Compared to CZTPA-1, CZTPA-2 has a higher hole mobility, which leads to good hole transport and enhances the charge transport in a planar PSC. CZTPA-2’s higher hole mobility can be attributed to its high hole transport capability. These results suggest that both a fast charge transfer and high hole transport capability contribute to a high PCE.

We tested the stability of the three HTM devices without encapsulation by storing them in the dark under air conditions for at least 400 h. The PCE over time curve was plotted and is shown in Figure 5b. The PCE of the CZTPA-2 device was still over 10%. In comparison, the PCE of Spiro-OMeTAD (dopant-free) device dropped below 10%. CZTPA-1 exhibited low performance, with 68.2% of the original PCE. CZTPA-2 and Spiro-OMeTAD (dopant-free) maintained 89.7% and 81.6% of their initial PCE, respectively. This test verified that the device based on CZTPA-2 has the best stability of the three HTMs. The long alkyl chain, which has good morphology, may influence the stability of a PSC device.

Finally, we also tested the capacitance of the three HTMs. The capacitance versus frequency was plotted and is shown in Figure 6. The capacitance is mainly caused by charge or ion accumulation at the perovskite interface, which leads to interfacial recombination. We can see the capacitance of the device based on CZTPA-2 is obviously smaller than that of CZTPA-1 and Spiro-oMeTAD (dopant-free), which confirms that the CZTPA-2 device has less interfacial recombination and a higher PCE.

## 4. Conclusions

Two new and low-cost hole transporting materials based on a carbazole core were designed and synthesized using a simple synthesis process. CZTPA-2 (dopant-free) achieved the best performance, with a PCE of 11.79%, a *J*_sc_ of 21.80 mA/cm^2^, a *V*_oc_ of 0.99 V, and a FF of 54.59%, which was slightly higher than that of Spiro-OMeTAD (dopant-free) and CZTPA-1 (dopant-free) and attributed to its higher hole transport mobility. The PL spectra, scanning electron microscopy images, and photoelectric properties indicate that CZTPA-2 with a longer alkyl chain has better optoelectrical properties. The CZTPA-2 (dopant-free) device also had the best stability, which remained at 89.7% of its original PCE after 400 h compared to CZTPA-1 and Spiro-OMeTAD (dopant-free). Besides this, its hole transport layer thickness is 35 nm. Therefore, we think that CZTPA-2 can also be used to modify an interface when compared to traditional HTMs (above 100 nm). The device based on CZTPA-2 exhibited good stability. We conclude that a longer alkyl chain may promote solubility and enhance the perovskite layer’s crystallinity.

## Figures and Tables

**Figure 1 nanomaterials-09-00935-f001:**
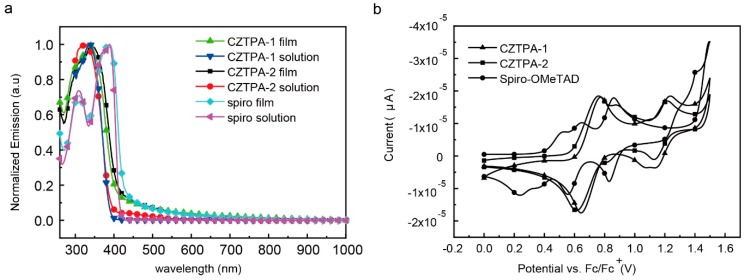
(**a**) Optical absorption spectra of 4,4′-(9-nonyl-9H-carbazole-3,6-diyl)bis (*N*,*N*-bis(4-methoxyphenyl)aniline) (CZTPA-1) and 4,4′-(9-methyl-9H-carbazole-3,6-diyl)bis (*N*,*N*-bis(4-methoxyphenyl)aniline) (CZTPA-2) in chloroform and thin films spin-coated from chloroform. (**b**) Cyclic voltammetry of CZTPA-1 and CZTPA-2.

**Figure 2 nanomaterials-09-00935-f002:**
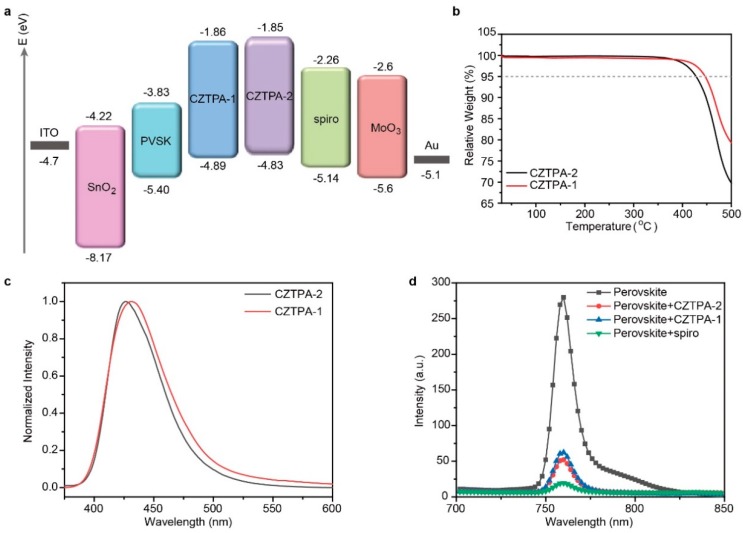
(**a**) Energy level diagram of a perovskite solar cell (PSC) with CZTPA-1, CZTPA-2, and 2,2′,7,7′-Tetrakis[*N*,*N*-di(4-methoxyphenyl)amino]-9,9′-spirobifluorene (Spiro-OMeTAD). (**b**) Thermogravimetric analysis (TGA) diagrams of CZTPA-1 and CZTPA-2. (**c**) Photoluminescence (PL) spectra of CZTPA-1 and CZTPA-2 thin film, excitation at 350 nm. (**d**) Photoluminescence spectra of perovskite, perovskite with Spiro-OMeTAD, perovskite with CZTPA-1, and perovskite with CZTPA-2, excitation at 500 nm.

**Figure 3 nanomaterials-09-00935-f003:**
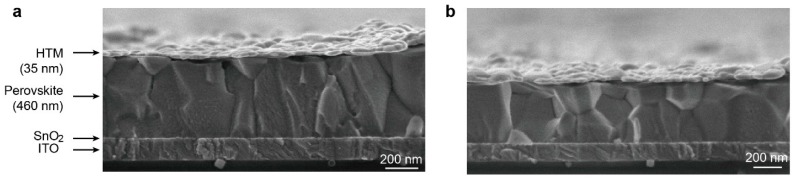
(**a**) A cross-sectional SEM image of the CZTPA-1 PSC. The scale bar is 200 nm. (**b**) A cross-sectional SEM image of the CZTPA-2 PSC. The scale bar is 200 nm.

**Figure 4 nanomaterials-09-00935-f004:**
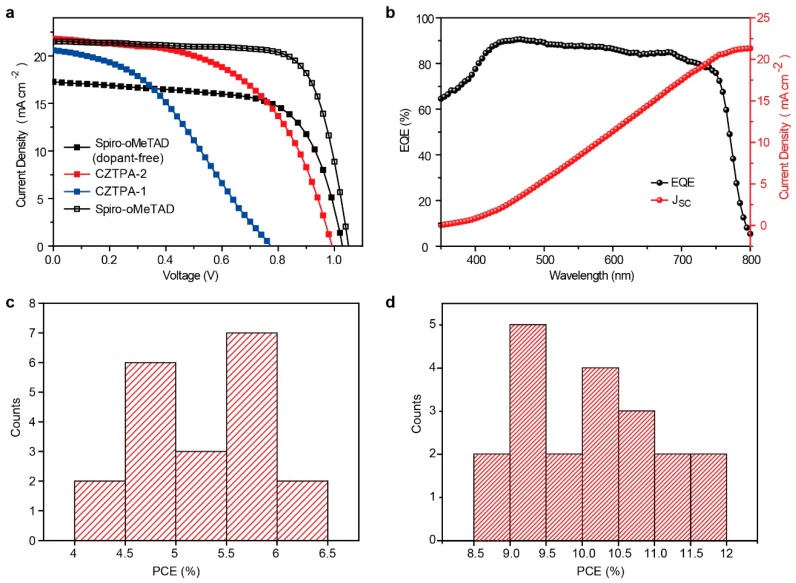
(**a**) *J–V* characteristics of PSCs based on the two hole transport materials (HTMs), the Spiro-OMeTAD, and the Spiro-OMeTAD (dopant-free). (**b**) External quantum efficiency (EQE) and *J*_sc_ spectra of PSCs with CZTPA-2. (**c**) Histograms of PCEs measured in 20 cells of CZTPA-1. (**d**) Histograms of PCEs measured in 20 cells of CZTPA-2.

**Figure 5 nanomaterials-09-00935-f005:**
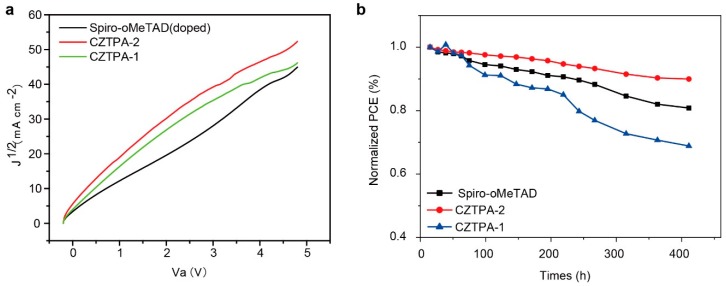
(**a**) Space-charge-limited current (SCLC) *J*^1/2^–*V* characteristics of CZTPA-1-, CZTPA-2-, and Spiro-OMeTAD-based hole-only devices measured in the dark. (**b**) Stability of CZTPA-1, CZTPA-2, and Spiro-oMeTAD (dopant-free) without encapsulation.

**Figure 6 nanomaterials-09-00935-f006:**
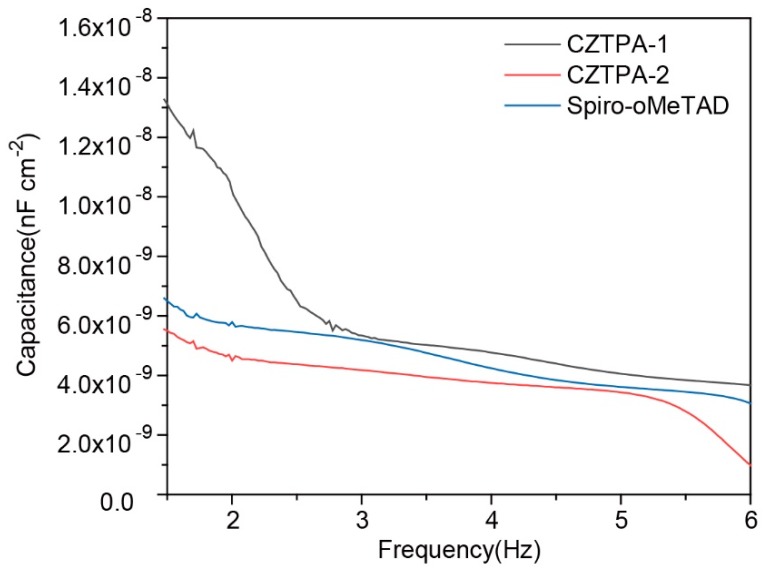
Capacitance versus frequency based on the devices of ITO/SnO_2_/perovskite/HTMs/Au.

**Table 1 nanomaterials-09-00935-t001:** Optical and electrochemical properties of CZTPA-1 and CZTPA-2.

	λ_max_ Sol (a) (nm)	λ_max_ Film (b) (nm)	λ_onset_ (nm)	*E*_g_^opt^ (c) (eV)	*E*_HOMO_ (eV)	*E*_LUMO_ (d) (eV)
CZTPA-1	335	337	409	3.03	−4.89	−1.86
CZTPA-2	327	339	416	2.98	−4.83	−1.85
Spiro-OMeTAD	386	390	428	2.90	−4.64	−1.74

(a) Maximum absorption peak in CH_2_Cl_3_ solution; (b) Maximum absorption peak of films on quartz glass; (c) Optical bandgap calculated from the absorption onset of films: *E*_g_^opt^ = 1240/λ_onset_ eV; (d) *E*_LUMO_ = *E*_HOMO_ + *E*_g_^opt^.

**Table 2 nanomaterials-09-00935-t002:** Photovoltaic data of PSCs based on the two HTMs, the Spiro-OMeTAD, and the Spiro-OMeTAD (dopant-free).

	*V*_OC_ (V)	*J*_SC_ (mA·cm^−2^)	FF (%)	PCE (%)	PCE Ave (%)
Spiro-OMeTAD (dopant-free)	1.02	17.26	66.14	11.74	10.02% ± 0.98
CZTPA-2	0.99	21.80	54.59	11.79	10.15% ± 0.90
CZTPA-1	0.77	20.58	38.01	6.05	5.27% ± 0.57
Spiro-OMeTAD	1.05	21.51	74.24	16.77	15.65% ± 0.71

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
