# Peer review of "Dopant-Free Hole Transport Materials with a Long Alkyl Chain for Stable Perovskite Solar Cells"

_nanomaterials, 2019, doi:10.3390/nano9070935_

Round 1
Reviewer 1 Report
In the article “Dopant-Free Hole Transport Materials with Long Alkyl Chain for Stable Perovskite Solar Cells,” Kai Wang and co-authors synthesized two types of organic molecules and used them as the hole-transporting material (HTM) in perovskite solar cells. The authors motivate the work by claiming a need to find an alternative to the typically used Spiro-OMeTAD. While the results have the potential to be interesting there are many issues that need to be addressed before I would recommend publication.
Issues:
· In lines 49-67 the discussion of the various HTMs used in perovskite solar cells is too shallow and only lists the names of compounds used. A figure/illustration of the compounds made by the authors and descriptions of why the various substitutions make a difference would improve this discussion greatly. Furthermore, the claim of easy and less expensive synthesis of the molecules used here should be more fully discussed and compared to Spiro.
· Lines 66-67: the sentence starting with “This may be attributed to the increased the alkyl chain…” is an incomplete sentence as it doesn’t say what was increased. If the authors mean that the length of the chain was increased and this led to increased charge mobility, then this statement is not intuitive as longer alkyl chains should have lower conductance (with everything else held constant).
· Figure 3 has no scale bar.
· Figure 5a the x-axis label “V-0.2” appears to be a typo.
· Line 189, the equation needs proper formatting as it is not possible to tell what operations are taking place.
· Line 191, what is the symbol “L”?
· Line 194, the statement “The high hole mobility leads to good hole transport” is wrong. The mobility in not high but in fact very low: two orders of magnitude lower than Spiro.
· Figure 6 does not add to the discussion and the statement of lower capacitance meaning fewer defects is an over-simplification and incorrect. More advanced impedance and admittance spectroscopy are needed to understand the role of CZTPA HTMs.
· The aging/stability tests are misleading and need clarification. First, it is written to sound like the 400h stability test was under operation not simply storage conditions which needs to be explicitly stated in the abstract. Secondly, what are the conditions during those 400h of storage other than “ambient”? i.e. what was the humidity, light levels, temperature, etc.?
· Many of the claims for better performance of CZTPA-2 vs. CZTPA-1 are based on the difference in chain length nonyl vs methyl but there are more factors that could determine the differences. To test this hypothesis, another chain length molecule should be tested.
Author Response
Dear Editor
Thank you for your email dated June 10, 2019 regarding the decision and two reviewers’ comments on our manuscript. We extremely hope that this work will be accepted in your journal. We truly understand the stringent standards required for publication in Nanomaterials. We appreciate the helpful comments and valuable suggestions provided by the two reviewers. For your quick check, I have highlighted our concerns in the First and Second reviewer’s report and also provide point-by-point responses as well as additional supporting information that may help you further evaluate the significance of our work.
Best regards
Hong Wang
South China University of Technology
Reviewer #1:
In lines 49-67 the discussion of the various HTMs used in perovskite solar cells is too shallow and only lists the names of compounds used. A figure/illustration of the compounds made by the authors and descriptions of why the various substitutions make a difference would improve this discussion greatly. Furthermore, the claim of easy and less expensive synthesis of the molecules used here should be more fully discussed and compared to Spiro.
Response: We are really grateful for the reviewer’s comments. So we added some supplements to the mentioned references.
In 2014, Sun and co-workers[27] have designed a series of HTMs based on carbazole, named x19 and x51. The x51 realizeding PCE of 9.8% due to the high charge-carrier mobility and conductivity than X19. Subsequently, Nazeeruddin and co-workers[28] studied bridged carbazole with biphenyl, by using silolothiophene as the bridge which obtained a PCE of 16.9%. They found compared to the spirofluorene linked triphenylamine HTMs, novel silolothiophene linked methoxy triphenylamines (Si-OMeTPAs) enable more stable PSCs. In the same year, Tang and co-workers[29] used carbazole as a core with four TPA as the side groups to achieve the PCE up to 18.32%. Recently, Khaja Nazeeruddin and co-workers[30] used anthra[1,2-b:4,3-b′:5,6-b′:8,7-b′′′]tetrathiophene as the core, by changinged the alkyl chain length of the methoxy groups on the triarylamine sites, , developing a series of materials. The device based on methyl substituted named ATT-OMe with the best PCE of 18.13% better than the device based on other longer alkyl chain HTMs. They believed the presence of alkyl chains decreasing the hole-transport properties. However, the performance of the PSCs based on carbazole is far from the classical Spiro-OMeTAD. Therefore, more efforts need to develop new small molecule HTMs matching with perovskite to instead of the Spiro-OMeTAD.
In this work, we developed two HTMs CZTPA-1 and CZTPA-2 based on TPA as the end group. They all have the advantage of easy synthesis steps and low cost. The two HTMs synthesis by one step Suzuki coupling. The raw material named 3,6-Dibromocarbazole with low cost ($0.3/g) plus 4-methoxy-N-(4-methoxyphenyl)-N-(4-(4,4,5,5-tetramethyl-1,3,2-dioxaborolan-2-yl)phenyl)aniline ($15/g) is clearly lower than Spiro-OMeTAD ($220/g). Notably, CZTPA-2 with the longer alkyl chain achieved better PCE of 11.79% with a short current density (Jsc) of 21.80 mA/cm2, an open circuit voltage (Voc) of 0.99 V and a fill factor (FF) of 54.59%. This may be attributed to the increased the alkyl chain on the carbazole improves the solubility of the HTMs in the devicecharge mobility of the device.
Price of Chemical $/g(mL) | Chemical dosage g(mL) | Synthesis cost $ | materials cost (one device) $ | |
Spiro-OMeTAD | 220 | 3.3x10-3 | 4 | |
1 | 0.3 | 2 | 0.6 | |
2 | 15 | 0.86 | 12.9 | |
C9H19Br | 0.25 | 1.53 | 0.3825 | |
KOH | 0.02 | 1.38 | 0.0276 | |
Pd[P(C6H5)3]4 | 6.32 | 1.15 x10-3 | 7.27 x10-3 | |
Tetrahydrofuran Extra Dry, with molecular sieves | 2.89 x10-2 | 20 | 0.578 | |
CZTPA-2 | 14.5 | 4.5x10-4 | 6.53 x10-3 |
Spiro-OMeTAD 73.2 mg/mL, one device 45μL
CZTPA-2 10 mg/mL, one device 45μL
Lines 66-67: the sentence starting with “This may be attributed to the increased the alkyl chain…” is an incomplete sentence as it doesn’t say what was increased. If the authors mean that the length of the chain was increased and this led to increased charge mobility, then this statement is not intuitive as longer alkyl chains should have lower conductance (with everything else held constant).
Response: We are really grateful for the reviewer’s comments. it may make mistake due to our lack of consideration. We found improvement of solubility is observed in the device of CZTPA-2 compared to CZTPA-1. The hole mobility of the two HTMs was been tested, the results shown CZTPA-2 own better hole mobility than CZTPA-1. But more importantly this is attributed to the significantly improved hole mobility of CZTPA-2, resulting in a significant increase in device efficiency. We change the statement.
“This is attributed to the significantly improved hole mobility of CZTPA-2, resulting in a significant increase in device efficiency.”
Figure 3 has no scale bar.
Response: We are really grateful for the reviewer’s comments. The scale bar is 200 nm. We have modified the relative section in Figure 3 in the revised manuscript.
Figure 5a the x-axis label “V-0.2” appears to be a typo.
Response: We are really grateful for the reviewer’s comments. We have modified the relative section in Figure 5a in the revised manuscript.
Line 189, the equation needs proper formatting as it is not possible to tell what operations are taking place.
Response: We sincerely appreciate the valuable comments from this reviewer. To make a better understanding to the hole transport, its formula is rewritten as follows: J = 9εrε0μVa2/8L3. We change the form of the formula to get the μ=8d3/9εrε0(J1/2/Va)2.
Line 191, what is the symbol “L”?
Response: We sincerely appreciate the careful review from this reviewer. The response to these questions is We sincerely apologize for our carelessness. The L should be replaced by the d. According to the equation, d is the thickness of the active layer.
Line 194, the statement “The high hole mobility leads to good hole transport” is wrong. The mobility in not high but in fact very low: two orders of magnitude lower than Spiro.
Response: The statement “The high hole mobility leads to good hole transport” is refer to this two HTMs compared to Spiro-OMeTAD (dopant-free). Compared to CZTPA-1,CZTPA-2 owns the high hole mobility which leads to good hole transport and , which enhances the charge transport in a planar PSC.
“L The d is 48 nm forCZTPA-1 and 56 nm for CZTPA-2, respectively.”
Figure 6 does not add to the discussion and the statement of lower capacitance meaning fewer defects is an over-simplification and incorrect. More advanced impedance and admittance spectroscopy are needed to understand the role of CZTPA HTMs.
Response: We really appreciate the reviewer’s valuable suggestions for improving the manuscript. The CZTPA-2 device owns a small capacitance. The capacitance is mainly caused by the charge or ion accumulation at the perovskite interface, which leads to interfacial recombination. So we think the low capacitance owns low interfacial recombination.
lower capacitance lead to fewer defects is modified to lower capacitance lead to low interfacial recombination.
The aging/stability tests are misleading and need clarification. First, it is written to sound like the 400h stability test was under operation not simply storage conditions which needs to be explicitly stated in the abstract. Secondly, what are the conditions during those 400h of storage other than “ambient”? i.e. what was the humidity, light levels, temperature, etc.?[楷1] [楷2]
Response: We sincerely appreciate the reviewer’s comments. The devices' storage environment was stored in the dark under air conditions without encapsulation with a humidity of 60% in the device stability test. Related information has been added to the article.
Many of the claims for better performance of CZTPA-2 vs. CZTPA-1 are based on the difference in chain length nonyl vs methyl but there are more factors that could determine the differences. To test this hypothesis, another chain length molecule should be tested.
Response: We truly understand the concern from the reviewer. According to our previous work, we have synthesized an HTM without alkyl chain substituted, which has poor dissolution in the chlorobenzene and its device exhibits terrible performance. Therefore, two molecules with different length of alkyl chain was designed. By comparing two HTMs, we found CZTPA-2 has the better performance due to its favorable solubility, better hole transport mobility and better morphology of perovskite/HTM film.

Reviewer 2 Report
This wok by Wang et al. describes two novel carbazole-based dopant-free HTMs for perovskite solar cells, and aims at evaluating the effect of alkyl chain lenght on performance and stability.
This topic is not new, and actually tons of papers with alternative HTMs are continuosly reported in literature. It is not clear what is the benefit of this new material with respect to the others, since it seems quite poorly performing. Comparing to undoped Spiro-OMeTAD is not very valuable, since it is known that undoped spiro is not working. Actually, I am surprised to see that the authors got such a high efficiency with this material. I do not recommend publication of this work, unless the authors can demonstrate the unique advantages of their materials and their approach
Minor comments:
- In Introduction, update the efficiency record for PSCs (Now 24.2%) in line 29. In line 28, replace solution preparation with "solution processability"
- Read through the manuscript again and fix typos and language issues. One example is in line 40 ", and was are"
- In order to have a fair comparison between the energy levels of the new material and those of spiro, please measure CV of spiro in same conditions, since big discrepancy is reported in literature about its HOMO/LUMO values
- In beginning os Results section, should there be a few lines about the design strategy of the new materials?
- Is thickness of the new materials in solar cells optimized? And what about roughness of the new materials films? AFM study would be needed to evaluate their morphology
- More details on the I-V measurements are needed: what has been the sweep rate? and what about the solar cell areas?
- Please report the standard deviations in Table 2
- Please present the stability data in a Figure (normalized efficiency vs. time) to improve readability
Author Response
Dear Editor
Thank you for your email dated June 10, 2019 regarding the decision and two reviewers’ comments on our manuscript. We extremely hope that this work will be accepted in your journal. We truly understand the stringent standards required for publication in Nanomaterials. We appreciate the helpful comments and valuable suggestions provided by the two reviewers. For your quick check, I have highlighted our concerns in the First and Second reviewer’s report and also provide point-by-point responses as well as additional supporting information that may help you further evaluate the significance of our work.
Best regards
Hong Wang
South China University of Technology
Reviewer #2:
I do not recommend publication of this work, unless the authors can demonstrate the unique advantages of their materials and their approach
Response: We truly understand the concern from the reviewer. Although the performance of these two materials is not as good as spiro, its hole transport layer thickness is only 35nm when it achieved the best PCE. Therefore, we think CZTPA-2 can be used as interface modification compared to traditional HTM (above 100 nm). Besides, these two HTMs have the advantage of easily modifying structure due to the N atom can be inserted with different groups. The cost for the synthesis of the HTMs is very competitive compared to spiro.
Price of Chemical $/g(mL) | Chemical dosage g(mL) | Synthesis cost $ | materials cost (one device) $ | |
Spiro-OMeTAD | 220 | 3.3x10-3 | 4 | |
1 | 0.3 | 2 | 0.6 | |
2 | 15 | 0.86 | 12.9 | |
C9H19Br | 0.25 | 1.53 | 0.3825 | |
KOH | 0.02 | 1.38 | 0.0276 | |
Pd[P(C6H5)3]4 | 6.32 | 1.15 x10-3 | 7.27 x10-3 | |
Tetrahydrofuran Extra Dry, with molecular sieves | 2.89 x10-2 | 20 | 0.578 | |
CZTPA-2 | 14.5 | 4.5x10-4 | 6.53 x10-3 |
Spiro-OMeTAD 73.2 mg/mL, one device 45μL
CZTPA-2 10 mg/mL, one device 45μL
In Introduction, update the efficiency record for PSCs (Now 24.2%) in line 29. In line 28, replace solution preparation with "solution processability"
Response: We sincerely appreciate the positive comments from the reviewer. We have updated new references in the manuscript.
“NREL Best Research Cell Efficiency Chart. (https://www.nrel.gov/pv/assets/pdfs/best-research-cell-efficiencies-190416.pdf)”
Read through the manuscript again and fix typos and language issues. One example is in line 40 ", and was are"
Response: We sincerely apologize for our carelessness. The typo and references format errors were corrected in the revised manuscript.
“The low cost, high stability and hole mobility are many advantages of HTMs based on inorganic materials.”
In order to have a fair comparison between the energy levels of the new material and those of spiro, please measure CV of spiro in same conditions, since big discrepancy is reported in literature about its HOMO/LUMO values
Response: As suggested by the reviewer, this latest change has been updated in the manuscript.
In beginning os Results section, should there be a few lines about the design strategy of the new materials?
Response: We thank the careful review from the reviewer. The related description has been added into the manuscript.
“The detailed experimental procedure was shown in SI. The design principles for our materials is improving planarity while increasing solubility. The carbazole group has good hole transporting ability and simple structures. By inserting N atom with a different alkyl chain can adjust optoelectronic properties and solubility. We choose 4-methoxy-N-(4-methoxyphenyl)-N-(4-(4,4,5,5-tetramethyl-1,3,2-dioxaborolan-2-yl)phenyl)aniline units by Suzuki coupling with carbazole core substituted by different alkyl chains. By Simple column chromatography separation and recrystallization, the aim materials can get.”
Is thickness of the new materials in solar cells optimized? And what about roughness of the new materials films? AFM study would be needed to evaluate their morphology
Response: Thanks a lot for having reviewed for our manuscript. We test the atomic force microscopy (AFM) images of two materials spin-coating onto perovskite films. Figures a and b represent the perovskite/CZTPA-1 and the perovskite/CZTPA-2 films, respectively. The roughness of both samples is slightly large, which may be due to the lower thickness of the HTMs. After comparison, it is found that the roughness of the film which HTMs is CZTPA-2 (RMS=20.968 nm) is significantly lower than that of CZTPA-1 (RMS=28.662 nm), it is attributed to the solubility of CZTPA-2 is higher so that could be better covered on the film and further improved the carrier transport of the device.
AFM images (5 μm x 5 μm) of CZTPA-2 (a) and CZTPA-1 (b) films.
More details on the I-V measurements are needed: what has been the sweep rate? and what about the solar cell areas?
Response: We thank the reviewer’s suggestion. The sweep rate of the measurements is all 0.02 V/s, the effective areas of solar cells are all 0.05 cm2. Related information has been added to the article.
Please report the standard deviations in Table 2
Response: We really appreciate the suggestions from this reviewer. We calculated the average PCE of CZTPA-2 devices with 10.15%±0.90,corresponding to CZTPA-2, CZTPA-1 devices own average PCE 5.27%±0.57. The average PCE of Spiro-OMeTAD (dopant-free) is 10.02%±0.98 and the corresponding PCE of Spiro-OMeTAD is 15.65%±0.71. Every data was based on 20 devices.
Please present the stability data in a Figure (normalized efficiency vs. time) to improve readability
Response: The reviewer’s comments are highly appreciated.
The stability data with (normalized efficiency vs. time) has updated in the blew. Same as previous stability tests, we plot the normalized efficiency versus times. The device based on CZTPA-2 has the best stability in three HTMs device in the glove box.

Round 2
Reviewer 2 Report
The revised version is greatly enhanced. Manuscript can be accepted for publication